# Can High-Frequency Intraoral Ultrasound Predict Histological Risk Factors in Oral Squamous Cell Carcinoma? A Preliminary Experience

**DOI:** 10.3390/cancers15174413

**Published:** 2023-09-04

**Authors:** Simone Caprioli, Giorgio-Gregory Giordano, Alessia Pennacchi, Valentina Campagnari, Andrea Iandelli, Giampiero Parrinello, Cristina Conforti, Riccardo Gili, Edoardo Giannini, Elisa Marabotto, Stefano Kayali, Bernardo Bianchi, Giorgio Peretti, Giuseppe Cittadini, Filippo Marchi

**Affiliations:** 1Radiology Unit, IRCCS Ospedale Policlinico San Martino,16132 Genova, Italy; simone.caprioli@hsanmartino.it (S.C.); cristina.conforti@hsanmartino.it (C.C.); giuseppe.cittadini@hsanmartino.it (G.C.); 2Department of Internal Medicine and Medical Specialties, University of Genova, 16100 Genova, Italy; giliriccardo.rg@gmail.com (R.G.); giorgio.peretti@hsanmartino.it (G.P.); filippomarchi@hotmail.it (F.M.); 3Otorhinolaryngology Unit, IRCCS Ospedale Policlinico San Martino, 16132 Genova, Italy; giojordan192@gmail.com (G.-G.G.); pennacchialessia96@gmail.com (A.P.); v.campagnari07@gmail.com (V.C.); giampiero.parrinello@hsanmartino.it (G.P.); 4Department of Surgical Science (DISC), University of Genova, 16100 Genova, Italy; 5Medical Oncology Unit, IRCCS Ospedale Policlinico San Martino, 16132 Genova, Italy; 6Gastroenterology Unit, Department of Internal Medicine, IRCCS Ospedale Policlinico San Martino, University of Genova, 16132 Genova, Italy; egiannini@unige.it (E.G.); elisa.marabotto@unige.it (E.M.); stefanokayali@gmail.com (S.K.); 7Department of Medicine and Surgery, University of Parma, 43121 Parma, Italy; 8Maxillo-Facial Surgery Unit, IRCCS Ospedale Policlinico San Martino, 16132 Genova, Italy; bernardo.bianchi@hsanmartino.it

**Keywords:** oral squamous cell carcinoma, intraoral ultrasound, Brandwein-Gensler score, histological risk factors

## Abstract

**Simple Summary:**

Oral cavity squamous cell carcinoma (OSCC) is a significant cancer burden worldwide. The current staging system for OSCC lacks accuracy in risk assessment: still, one-third of patients affected by stage I and II OSCC can develop locoregional recurrences following adequate treatment. Researchers explored the use of high-frequency intraoral ultrasonography (IOUS) to predict histological risk factors in OSCC. The results demonstrated that IOUS accurately measured the depth of invasion (DOI) and showed a strong correlation with adverse histopathological features (APFs). This promising new imaging technique could improve risk stratification for OSCC patients, potentially leading to better treatment outcomes.

**Abstract:**

Despite advancements in multidisciplinary care, oncologic outcomes of oral cavity squamous cell carcinoma (OSCC) have not substantially improved: still, one-third of patients affected by stage I and II can develop locoregional recurrences. Imaging plays a pivotal role in preoperative staging of OSCC, providing depth of invasion (DOI) measurements. However, locoregional recurrences have a strong association with adverse histopathological factors not included in the staging system, and any imaging features linked to them have been lacking. In this study, the possibility to predict histological risk factors in OSCC with high-frequency intraoral ultrasonography (IOUS) was evaluated. Thirty-four patients were enrolled. The agreement between ultrasonographic and pathological DOI was evaluated, and ultrasonographic margins’ appearance was compared to the Brandwein-Gensler score and the worst pattern of invasion (WPOI). Excellent agreement between ultrasonographic and pathological DOI was found (mean difference: 0.2 mm). A significant relationship was found between ultrasonographic morphology of the front of infiltration and both Brandwein-Gensler score ≥ 3 (*p* < 0.0001) and WPOI ≥4 (*p* = 0.0001). Sensitivity, specificity, positive predictive value, and negative predictive value for the IOUS to predict a Brandwein-Gensler score ≥3 were 93.33%, 89.47%, 87.50%, and 94.44%, respectively. The present study demonstrated the promising role of IOUS in aiding risk stratification for OSCC patients.

## 1. Introduction

Oral cavity squamous cell carcinoma (OSCC) represents a significant global cancer burden, accounting for over fifty thousand new diagnoses per year in the United States [1], with an estimated cancer-specific mortality of around 30% [2,3,4,5]. The tongue is affected in up to 50% of all cases [6], demonstrating a tendency for local tissue and regional lymphatic spread early in the course of the disease.

As of today, a few issues in the management of OSCC can be disclosed. Regardless of the progresses made in surgical techniques and multidisciplinary care of OSCC in the last two decades, the oncologic outcomes of patients affected by OSCC have not substantially improved [7,8]. The Union for International Cancer Control (UICC)/American Joint Committee on Cancer (AJCC) 8th edition Tumor Node Metastasis (TNM) staging systems for oral cancer remain inadequate for accurate risk stratification [9,10], even though a step forward was made by introducing the depth of invasion (DOI) into the staging system. DOI has shown to be a more accurate independent prognostic factor for local recurrence, subclinical nodal metastasis, and survival than tumor thickness [11,12]. Therefore, having a precise preoperative measurement of DOI allows experts to accurately plan the resection and a possible prophylactic neck dissection (recommended by several experts with a DOI > 4 mm) [13,14,15].

Currently, one-third of patients affected by stage I and II OSCC can develop locoregional recurrences following adequate treatment according to current guidelines [16]. Locoregional recurrences are demonstrated to have a strong link to adverse histopathological features (APFs) that are not yet included in the staging system, such as lymphovascular invasion (LVI), perineural invasion (PNI), and the worst pattern of invasion (WPOI) [17]. Additionally, oncological outcomes and prognosis of stage III and IV OSCC are extremely heterogeneous due to the absence of subgrouping in patients with and without nodal metastasis.

Balasubramanian and colleagues described nomograms based on the latest UICC/AJCC staging system, including many relevant APFs, such as maximal dimension of the tumor, grading, PNI, LVI, status of surgical margins, DOI, bone invasion, pathological T (pT) category, pathological N (pN) category, maximum dimension of metastatic lymph nodes, and evidence of extranodal extension (ENE) [18].

In order to introduce an adequate predictive model, many histological grading systems have been described in recent times [19]. In 2005, Brandwein-Gensler et al. proposed a histological risk assessment model claiming to have a superior prognostic value than other systems described in the literature [20]. This model considers three histological parameters: WPOI, lymphocytic host response (LHR), and PNI. By taking PNI and LHR into account, along with WPOI, a more complete assessment of tumor environment can be achieved. The score ranges from 0 to 9, creating three risk groups [20]. Such model showed a significant correlation between locoregional recurrence and overall survival, especially in early-stage OSCC [21].

The capability to identify preoperatively aggressive features of OSCC would allow the surgeon to predict the biological behavior and thus tailor the tumor excision or the neck treatment, especially for patients in the early stage, among whom the histopathological heterogeneity causes wide dissimilarities in outcome [22]. Every surgeon would seek prior knowledge of whether the tumor possesses aggressive histological characteristics that necessitate a more extensive resection during the operation, which is the only prognostic factor dependent on the operator [23]. Thus, a recent study revealed that different WPOIs determine the ideal extent of surgical margins as 1.7 mm for patients with types 1–3 and 7.8 mm in patients with types 4/5 [24].

Computed tomography (CT) and Magnetic Resonance Imaging (MRI) are the radiological techniques of choice for evaluating the extension of OSCC, especially MRI, which is widely used even in smaller lesions [25]. Conversely, the use of ultrasound (US) has historically been limited only to the evaluation of the neck, thyroid gland, or salivary glands and was only recently applied to OSCC [26]. The introduction of the DOI and the development of new ultrasonographic technologies has reignited interest in this radiological technique [14].

Recently, multiple authors have been trying to identify which radiological examination provides the most precise preoperative evaluation of DOI, showing that high-frequency intraoral ultrasonography (IOUS) may play a promising role thanks to its excellent spatial resolution [14,15]. However, to the authors’ knowledge, there is no research study that has ever attempted to predict the histological risk factors in OSCC with radiological tools. We aim to investigate whether IOUS may be used for predicting histological risk factors in OSCC. Our primary endpoint is to investigate the capability of IOUS to predict a Brandwein-Gensler score ≥3 and WPOI ≥4; our secondary endpoint is confirming the accuracy of IOUS in the DOI measurement.

## 2. Materials and Methods

### 2.1. Patients

Patients with biopsy-proven OSCC who underwent IOUS from April 2021 to July 2023 were included in this study. Inclusion criteria were pre-surgical IOUS evaluation, pathologically proven OSCC, complete surgical excision, and evaluation of histological risk factors by assessing the Brandwein-Gensler score. Every patient underwent IOUS and then an incisional biopsy performed by head and neck surgeons. In the case of biopsy-proven OSCC, the patient was enrolled and surgically treated. Exclusion criteria were neoadjuvant radiotherapy or chemotherapy, benign lesions, in situ squamous cell carcinoma, and non-squamous cell carcinoma tumors.

### 2.2. Intraoral Ultrasound and Image Examination

Intraoral ultrasounds were performed before biopsy by a dedicated head and neck radiologist using a high-frequency “hockey stick” probe (22–28 MHz), coated with a sterile latex cover, in which a small amount of ultrasound gel was placed. The presence of the ultrasound gel inside the cover created a distance between the probe and the mucosa; this expedient enabled the operator to clearly depict exophytic or ulcerated lesions [27]. The operator applied gentle pressure in order to limit compression distortion of the thickness and morphology of the examined lesion. Ultrasonographic depth of invasion and morphology of the tumoral front of infiltration (FOI) were evaluated. In the existing literature, patterns of invasion are categorized into aggressive (WPOI 4–5) and non-aggressive (WPOI 1-2-3). However, these features are revealed only through definitive pathological examination. Consequently, we hypothesized that an invasive growth type observed on the ultrasound corresponds to an aggressive pattern, whereas a pushing growth type corresponds to a non-aggressive one. Moreover, depth of invasion was measured, as suggested by the NCCN Guidelines [28], from the mucosal surface to the deepest point of infiltration along an ideal perpendicular line, excluding exophytic parts and including ulcerated portions of the lesions. Imaging features were retrospectively analyzed on static anonymized images using the institutional Picture Archival and Communication System (PACS). Morphology of neoplastic FOI was defined as “regular” if smooth margins were observed and “irregular” if lobulated, spiculated, or indistinct margins were observed on IOUS (Figure 1).

### 2.3. Statistics

Statistical analysis was performed using MedCalc Software 22.009. Fisher’s exact test was used to evaluate the relation between margin appearance on IOUS and a Brandwein-Gensler score ≥ 3. Fisher’s exact test was also applied to assess the relation between margin appearance and WPOI ≥ 4. Specificity, sensitivity, positive predictive values, and negative predictive values were calculated for margins’ appearance on the US to predict a Brandwein-Gensler score > 3. The agreement between pathological (pDOI) and radiological depth of invasion (usDOI) was assessed with a Bland–Altman plot.

## 3. Results

A total of 34 patients met the inclusion criteria. Clinical and demographic data are presented in Table 1. The primary site was the tongue in 26 patients. Within this category, 24 patients were affected by lateral tongue surface carcinoma and 2 by dorsal tongue surface carcinoma. The floor of the mouth was affected in 4 patients, while the alveolar crest was affected in 4 patients. On histology, T-stage was pT1 for 11 lesions, pT2 for 11 lesions, pT3 for 8 lesions, and pT4a for 4 lesions. Eight patients had nodal metastasis at the time of diagnosis. The Brandwein-Gensler score was high (≥3) in 16 patients and low or intermediate (<3) in 18 patients. The Bland–Altman plot demonstrated excellent agreement between the usDOI and pDOI, although a mean systematic error was observed between the two (0.2 mm) (Figure 2). FOI was categorized as “irregular” in 14 patients with a Brandwein-Gensler score ≥ 3 and in 1 patient with Brandwein-Gensler score < 3, while it was classified as “regular” in 2 patients with a Brandwein-Gensler score ≥ 3 and in 17 with a Brandwein-Gensler score < 3 (Figure 3). A significant relationship between the Brandwein-Gensler score and margin appearance on the ultrasound was observed (Fisher’s exact test, *p* < 0.0001). FOI was designated as “irregular” in 14 patients with WPOI ≥ 4 and in 1 patient with WPOI < 4, while FOI was labeled as regular in 14 patients with WPOI < 4 and in 5 patients with WPOI ≥ 4 (Figure 4). A significant relationship between WPOI ≥ 4 and margin appearance on the ultrasound was found (Fisher’s exact test, *p* = 0.0001). Sensitivity, specificity, positive predictive value, and negative predictive value for IOUS in predicting a Brandwein-Gensler score ≥ 3 were 93.33%, 89.47%, 87.50%, and 94.44%, respectively.

## 4. Discussion

An estimated 66,650 new cases of oral cancer are diagnosed in Europe every year. Compared to other sites of the head and neck, the oral cavity permits easier access, also allowing for self-examination. However, the diagnosis often occurs in an advanced stage. An accurate investigation of all the upper aero-digestive tracts is always indicated because the risk factors for tumors of the oral cavity are the same as in other locations such as the oropharynx, larynx, and hypopharynx. Along with fibroscopy, a palpatory examination is still essential for both identifying the primary lesion and searching for suspicious swellings in the neck. Despite innovations in treatment, survival has not significantly improved over the years for patients with OSCC. Nodal metastases and recurrences are not rare, even at earlier stages. Negative outcomes are strongly associated with APFs, such asLVI, PNI, and theWPOI, which is not yet included in the staging system. A recently described prognostic scoring system was proposed by Brandwein-Gensler et al. using these adverse histological features [20]. Among them, WPOI has been demonstrated to represent a consistent and detrimental adverse prognostic factor. Intuitively, tumors invading with a discohesive and dispersed growth pattern are more aggressive and develop more multifocal and recurrent tumors than those growing in a bulky and pushing fashion [29]. The presence of a high WPOI score is a significant independent prognostic factor, resulting in poor survival outcomes in OSCC. WPOI is associated with a higher rate of nodal metastases, PNI, and LVI, either intratumoral or extratumoral [30]. Moreover, Yue et al identified a correlation between the WPOI of the surrounding soft tissue and the tumoral abutment to the bone, demonstrating a significant association with infiltrative mandibular invasion [31]. Several studies reported how cancer cells from WPOI-5 tissues have dysregulated cell-to-cell adhering signals and cellular movement pathways for tumor invasion [32]. Thus, the tumor invasive front and the tumor/stroma interface constitute a dynamic microenvironment. Tumorigenic and metastatic stages are not solely dependent on genetic and epigenetic alternations; they might represent the result of altered organ homeostasis rather than the dysregulated control of the tumor cell. Tumor–stromal interactions significantly influence metastasis [24,33]. In a recent study, Köhler et al. demonstrated that the ideal extent of surgical margins could be dramatically affected based on different scores of WPOI, ranging from the conventional 5mm to 1.7 mm for patients with type 1/2/3 up to a wide 7.8 mm in the case of WPOI 4 or 5 [30]. Incorporating this valuable information in a preoperative setting could heavily influence the surgical plan. The unanimous and detrimental relevance of WPOI raises the question of whether this parameter should be included in the guidelines as an indicator to address patients to adjuvant treatment [34]. In fact, it exhibits a constant and incremental relationship with the DOI, another well-recognized APF [35]. Conversely, when DOI is the sole APF, it is not able to significantly impact survival. We might surmise that deterioration in prognosis with increasing DOI largely reflects an association with other pathological risk factors and consider DOI mainly as a by-product of WPOI [36]. DOI likely reflects not only underlying disease biology but also delayed patient presentation at diagnosis. Thus, a subset of patients with thicker primary tumors may exhibit relatively biologically indolent disease but still present with high DOI. The feasibility of predicting WPOI preoperatively in the biopsy specimen has been assessed by Pu et al. [37]. Despite promising results, the authors did not obtain an adequate overlap between the intermediate biopsy pattern and the final WPOI type 4 and 5, rendering the biopsy pattern unreliable for preoperative prognosis prediction. Several efforts have been made to anticipate other APFs before surgery. However, information provided by the biopsy concerning unfavorable histologic parameters in preoperative biopsy specimens showed poor correlation with the subsequent resection specimen. In early OSCC, the differentiation grade revealed by the biopsy demonstrated limited predictive power for the grading of the resection specimen. Moreover, it could not be correlated with the presence of nodal metastasis and appears to lack prognostic value concerning outcomes [38]. Similarly, regarding PNI and LVI, the preoperative biopsy specimens did not accurately represent the final post-surgical specimen and the subsequent risk of occult metastasis [39]. New emerging indicators for biological behavior have been explored recently. Tongue depapillation, revealed during endoscopic examination, has been shown to represent a reliable surrogate of PNI [40]. However, this feature only applies to the tongue dorsum and border, where papillae are located. These studies are preliminary and require broader validation by other tertiary centers [41]. As mentioned earlier, imaging plays a crucial role in accurately defining the extension of the tumor and identifying nodal and distant metastasis, especially in advanced tumors. Magnetic resonance is considered the preferred imaging technique for OSCC staging, due to its excellent soft tissue resolution, while computed tomography (CT) can provide complementary information about cortical bone erosions [25]. Ultrasound is a valuable, non-invasive method that allows for the evaluation of various regions of the body, including the head and neck area, where it serves as the primary examination for neck swelling. Offering highly detailed anatomical information, it provides an excellent assessment of the superficial anatomy of the neck [42], as well as evaluation of salivary glands and the thyroid gland [43]. However, it is generally considered less informative when it comes to the evaluation of oral cancer, due to the lack of acoustic windows in the suprahyoid neck and inability to assess bone.

While intraoral ultrasound (IOUS) might be considered a rather unusual application, interest in its use for OSCC staging is growing exponentially. Shintani and colleagues were the first to describe the use of US applied to tumors of the oral cavity for measuring tumor thickness (at that time, the concept of DOI applied to OSCC was unknown) [26]. Many subsequent experiences followed this first paper [33,44,45,46,47,48,49,50,51,52,53]. Several studies in the literature have compared ultrasonographical DOI to other methods (MRI and CT). A recent study by our group showed slightly better results for IOUS than MRI, although a mean systematic error was observed between ultrasonographic DOI and pathological DOI (mean difference: 0.7 mm), with high sensitivity, specificity, positive predictive value, and negative predictive value in determining invasion of subepithelial connective tissue [54]. Similar results were obtained by Rocchetti et al [55]. Takamura and colleagues also demonstrated a higher reliability in determining DOI for T1 and T2 OSCCs using IOUS compared to MRI, with a good radiological–pathological agreement (mean difference between ultrasonographic DOI and pathological DOI: 0.2 mm) [56]. Recently, Izzetti et al. evaluated ultra-high-frequency ultrasound (70 MHz linear probe) for determining tumor thickness and DOI, with good results [57]. Nevertheless, results are still controversial because IOUS has been used more frequently to measure tumor thickness than DOI. Metanalysis and systematic reviews in the literature present contradictory findings [15,58,59]. However, as previously stated, even though DOI is one of the two key points for determining the T-stage of OSCC, its link with an aggressive disease is debated. A bulky tumor with a thicker DOI at presentation may only reflect a delayed diagnosis rather than an aggressive tumor, as a subset of these patients might have a biologically indolent disease. Moreover, DOI is unable to independently affect survival when it is the sole APF. With that said, a reliable method to preoperatively estimate OSCC aggressiveness is desirable. Among radiological techniques, high-frequency US seems to be particularly suited for this purpose, thanks to its intrinsic characteristics. As a matter of fact, US has the highest spatial resolution, as mentioned previously. Spatial resolution is defined as the ability of an imaging system to detect structures that are closely located and to discriminate between them. Thanks to this characteristic, if the probe is placed onto an oral lesion, FOI can be clearly detected and regular or irregular tumoral margins can be easily distinguished. Our study aimed to investigate whether irregular margins detected on an IOUS might be a predictor of a more invasive pattern of growth. As a preliminary phase, as previously conducted in former studies [54], agreement between usDOI and pDOI was evaluated. These data were used as an indirect confirmation that ultrasound-detected FOI corresponded with the actual tumor border.

Excellent agreement between usDOI and pDOI was verified. Thus, the morphology of FOI was evaluated. We considered the histological risk score proposed by Brandwein-Gensler et al., as it is widely used at our center, as well as WPOI because of its independent prognostic significance [20,30]. However, due to the small sample, we only tested the “high-risk” category of the Brandwein-Gensler system and the more aggressive scores of WPOI (WPOI 4 and 5). An irregular FOI was statistically associated with a higher risk category according to the Brandwein-Gensler score (Fisher’s exact test, *p* < 0.0001). Hypothetically, an irregular shape of the deep tumor margin may correspond to a more “infiltrative” rather than “pushing” pattern of growth. Moreover, WPOI is a histological feature of a discohesive and dispersed growth pattern, and as a consequence, indicates a more aggressive growth. Irregular FOI was strongly associated with WPOI 4/5 in our series (Fisher’s exact test, *p* = 0.0001). Because WPOI as well as PNI, two of the three features of the Brandwein-Gensler score, are associated with an increased risk of recurrence [60], an irregular FOI may be considered an imaging feature of aggressiveness and, theoretically, a risk factor for recurrence. To the authors’ knowledge, this is the first study that attempted to analyze tumoral margins detected on IOUS for predicting histological risk factors. The results are promising, although some important limitations must be noted. Firstly, the retrospective nature of the study may be attributed to potential selection biases. Moreover, the small number of patients, mainly at an early stage and with a relatively short follow-up, did not allow authors to evaluate the correlation between irregular FOI and the onset of lymph node metastases or recurrence. Additionally, a larger sample could enable a more specific classification of FOI morphology, distinguishing, for instance, smooth, lobulated, spiculated, or indistinct margins and comparing these categories to the five different classes of WPOI. However, this study should be considered a preliminary experience, and further investigations with a larger sample size are needed for validating and confirming results.

## 5. Conclusions

High-frequency intraoral ultrasonographic examination of the oral mucosa is a method that remains largely unfamiliar in most centers; yet, it has shown very promising results, as demonstrated by our study and corroborated by the existing literature. In our retrospective series, we found a strong association between FOI and high-risk histological features, such as the Brandwein-Gensler score and WPOI. However, despite the strong association between the type of FOI and Brandwein-Gensler histological risk, further studies should be conducted to validate our results. Many predictors of prognosis are well known in OSCC, but only one is in our hand, which is the status of margins. A more aggressive WPOI demands a wider margin. However, identification of these patterns in the pre-operative biopsy has shown to be unreliable. Some new technologies, such as intraoperative Raman spectroscopy and narrow band imaging-guided surgical margins, could increase the likelihood of successful clear margins in surgery. However, these technologies are not available in most treatment centers worldwide and their cost-effectiveness ratio is still undetermined. Our study revealed a powerful tool in pre-operative identification of WPOI. Although promising, the current state of knowledge suggests that the use of IOUS in the evaluation of the tumor FOI does not significantly alter the therapeutic approach of tumors of the oral cavity, which continues to rely predominantly on surgical intervention, with the aim of obtaining 5–10 mm negative margins. In conclusion, our experience provides a preliminary answer to the need to create a reliable, non-invasive, cost-effective tool for predicting histological adverse features before surgery. In the future, head and neck cancer specialists will need to possess the ability and the tools to adjust the extent of resection according to radiological parameters that can serve as indicators of the underlying adverse pathological characteristics.

## Figures and Tables

**Figure 1 cancers-15-04413-f001:**
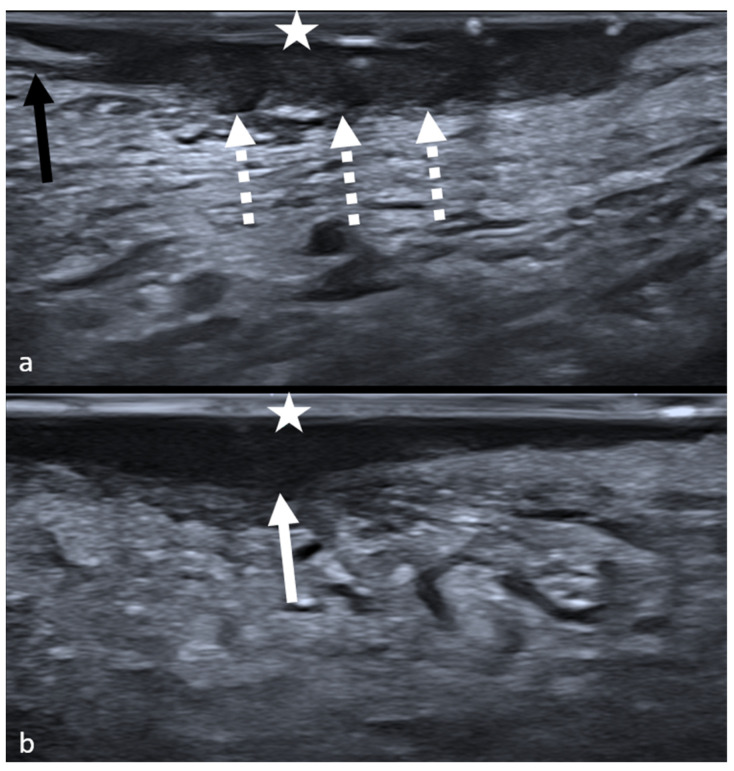
Intraoral ultrasound of two oral squamous cell carcinomas. (**a**) Irregular margins: dotted arrows indicate spiculae and lobulations of tumoral front of infiltration. Black arrow: superior longitudinal muscle, infiltrated by the lesion. White star: ultrasound gel. (**b**) Regular margins: smooth, regular front of infiltration (white arrow). White star: ultrasound gel.

**Figure 2 cancers-15-04413-f002:**
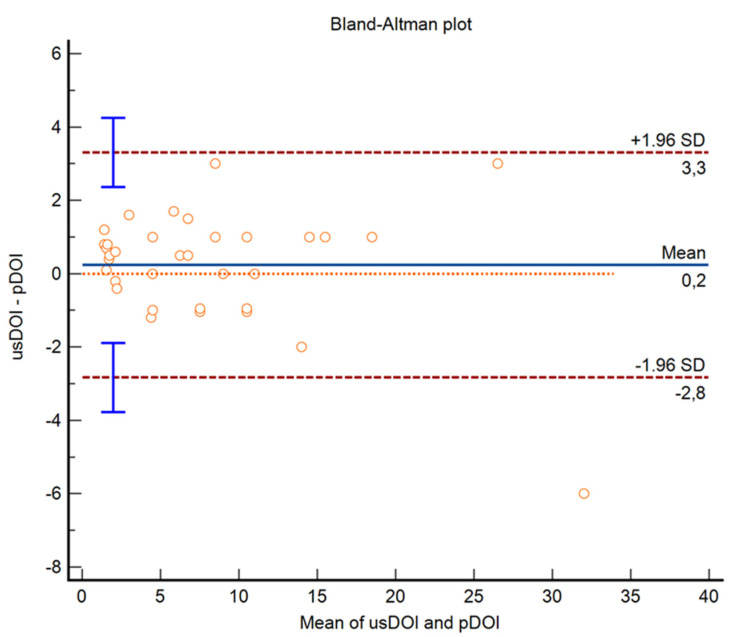
Bland–Altman plot comparing ultrasonographic depth of invasion (usDOI) and pathological depth of invasion (pDOI). Only one measure was outside the limits of agreement. Mean difference between usDOI and pDOI was 0.2 mm. Orange dotted line: line of equality (difference = 0); blue line: mean difference; red dashed line: limits of agreement; vertical blue line: 95% confidence interval of limits of agreement.

**Figure 3 cancers-15-04413-f003:**
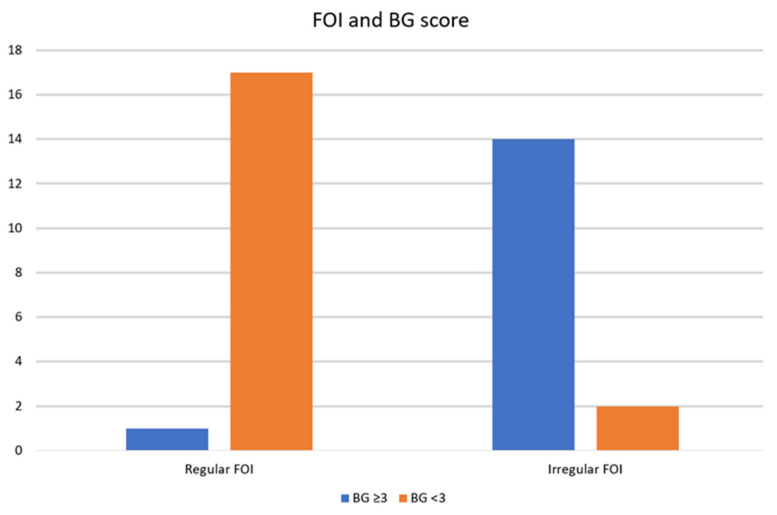
Frequency charts comparing regular or irregular front of infiltration and Brandwein-Gensler score (FOI: front of infiltration; BG: Brandwein-Gensler score).

**Figure 4 cancers-15-04413-f004:**
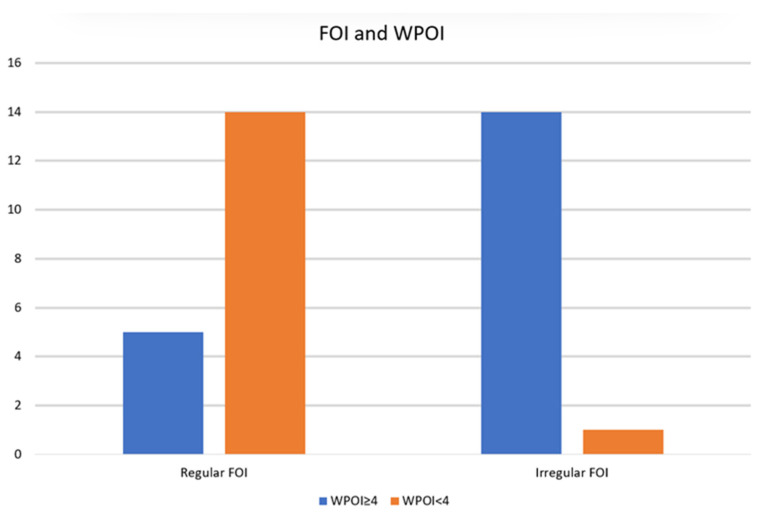
Frequency charts comparing regular or irregular front of infiltration and worst pattern of invasion (FOI: front of infiltration; WPOI: worst pattern of invasion).

**Table 1 cancers-15-04413-t001:** Demographic and clinical data of patients included in the study.

	N
Patients	34
Female	21 (61.76%)
Average age (standard deviation)	65 (17.15)
pT1	11 (32.35%)
pT2	11 (32.35%)
pT3	8 (23.53%)
pT4a	4 (11.76%)
Lymph node(s) metastasis	8 (23.52%)
Brandwein-Gensler score ≥ 3	16 (47.06%)

## Data Availability

Not applicable.

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
