# Peer review of "Can High-Frequency Intraoral Ultrasound Predict Histological Risk Factors in Oral Squamous Cell Carcinoma? A Preliminary Experience"

_cancers, 2023, doi:10.3390/cancers15174413_

Round 1
Reviewer 1 Report
Dear Authors,
The idea behind the article is of absolute interest. However, it is not the first paper to deal with the topic of ultrasound DOI. I would therefore suggest in the introduction to revise the sentence (However, to authors' knowledge, there is no research that ever tried to predict histological risk factor in OSCC with radiological tools). There are several articles that have addressed this issue. Among them:
1. Rocchetti F, Tenore G, Montori A, Cassoni A, Cantisani V, Di Segni M, Di Gioia CRT, Carletti R, Valentini V, Polimeni A, Romeo U. Preoperative evaluation of tumor depth of invasion in oral squamous cell carcinoma with intraoral ultrasonography: a retrospective study. Oral Surg Oral Med Oral Pathol Oral Radiol. 2021 Jan;131(1):130-138. doi: 10.1016/j.oooo.2020.07.003. Epub 2020 Jul 9. PMID: 32792295. - Which you also cited, by the way;
2. Izzetti R, Nisi M, Gennai S, Oranges T, Crocetti L, Caramella D, Graziani F. Evaluation of Depth of Invasion in Oral Squamous Cell Carcinoma with Ultra-High Frequency Ultrasound: A Preliminary Study. Applied Sciences. 2021;11(16):7647. https://doi.org/10.3390/app11167647
The title is catchy, but the study design is lacking, as well as in the body of the text.
Regarding the materials and methods section this is severely incomplete. It lacks: in the selection of patients, the calculation of a suitable sample size. Who performed the biopsy? What time was the ultrasound performed, before or after the biopsy incision? how was the excised specimen stored?
Also, the method used of "resting" the probe on the lesion, in the case of superficial lesions, risks losing some important information(Di Stasio D, Montella M, Romano A, Colella G, Serpico R, Lucchese A. High-Definition Ultrasound Characterization of Squamous Carcinoma of the Tongue: A Descriptive Observational Study. Cancers. 2022;14(3):564. https://doi.org/10.3390/cancers14030564). The authors posed the question of what was the healthy tissue from which to draw the DOI surface line? Only 0.2 mm difference between pDOI and usDOI are reported in the results, how is this possible? One has to consider that the excised piece undergoes a number of dimensional changes from the microscopic collection, ranging from 10 to 20%.
In short, the work has some issues that jump out at those who are experts on the subject and therefore needs major revisions. Translated with www.DeepL.com/Translator (free version)
Author Response
Dear Reviewer,
We sincerely appreciate your insightful revisions and astute suggestions. The requested changes have been successfully incorporated. We would like to clarify that the primary endpoint of our work is not evaluating the radiological depth of invasion (DOI) by ultrasound, which is a secondary endopint. Instead, our primary objective is assessing the pattern and the front of invasion by ultrasound (FOI). Moreover, from the NCCN guidelines version 2.2023 “ Adverse pathologic features” are: “extranodal extension, positive margins, close margins, pT3 or pT4 primary, pN2 or pN3 nodal disease, nodal disease in levels IV or V, perineural invasion, vascular invasion, and lymphatic invasion”, not DOI. From the AJCC Cancer Staging Manual, Eighth Edition (2017) “DOI is a staging criteria”. Thus, in the title we mentioned “predict histological risk factors”, not "predict DOI". We aim to investigate the capability of US to predict pathological risk factors not included in staging system such as worst pattern of invasion (WPOI), that still have a strong prognostic value. In the introduction we mentioned in line 66 “ Locoregional recurrences are demonstrated to have a strong link to adverse histopathological features that are still not included in the staging system, such as lymphovascular invasion (LVI), perineural invasion (PNI) and worst pattern of invasion (WPOI)“. We are well aware that the radiological DOI has already been thoroughly investigated by numerous studies, including our own Center's pioneering efforts, references:
1) Caprioli S, Casaleggio A, Tagliafico AS, Conforti C, Borda F, Fiannacca M, Filauro M, Iandelli A, Marchi F, Parrinello G, Peretti G, Cittadini G. High-Frequency Intraoral Ultrasound for Preoperative Assessment of Depth of Invasion for Early Tongue Squamous Cell Carcinoma: Radiological-Pathological Correlations. Int J Environ Res Public Health. 2022 Nov 12;19(22):14900. doi: 10.3390/ijerph192214900. PMID: 36429617; PMCID: PMC9690087.;
2) Marchi F, Filauro M, Iandelli A, Carobbio ALC, Mazzola F, Santori G, Parrinello G, Canevari FRM, Piazza C, Peretti G. Magnetic Resonance vs. Intraoral Ultrasonography in the Preoperative Assessment of Oral Squamous Cell Carcinoma: A Systematic Review and Meta-Analysis. Front Oncol. 2020 Feb 4;9:1571. doi: 10.3389/fonc.2019.01571. PMID: 32117789; PMCID: PMC7010633.;
Since several authors unveiled the power of ultrasound in DOI evaluation, we aimed to investigate its ability to predict different pattern of aggressiveness. However, all the features named before are revealed only by definitive pathological examination. Thus, we hypothesized that an invasive type of growth on ultrasound reflects an aggressive pattern, while a pushing type of growth a non-aggressive pattern. Our results confirmed the hypothesis in our series. Our secondary endpoint is confirming the accuracy of ultrasound in DOI measurement, which has already been reported; moreover, as we stated in line 298, agreement between radiological DOI and pathological DOI was used in our study as an indirect confirmation that ultrasound-detected FOI corresponded with the actual tumor border.
Regarding the biopsy timing:
-every patients underwent ultrasound, and then incisional biopsy performed by the head and neck surgeon. In case of positive biopsy for SCC, the patients was enrolled and surgically treated. Text has been changed.
-is of common practice in our Center, as in every tertiary Center, to perform a biopsy following imaging studies, to avoid artifacts. Text has been changed.
Upon scrutinizing the results of the correlation between radiological and pathological measurements of depth of invasion (DOI), we also experienced genuine excitement. This can be partially ascribed to the fact that, as the reviewer can notice from the provided references (Filauro M, Missale F, Marchi F, Iandelli A, Carobbio ALC, Mazzola F, Parrinello G, Barabino E, Cittadini G, Farina D, Piazza C, Peretti G. Intraoral ultrasonography in the assessment of DOI in oral cavity squamous cell carcinoma: a comparison with magnetic resonance and histopathology. Eur Arch Otorhinolaryngol. 2021 Aug;278(8):2943-2952. doi: 10.1007/s00405-020-06421-w. Epub 2020 Oct 21. PMID: 33084951; PMCID: PMC8266699.), such a methodology has been in practice at our institution for approximately 7 years. This longevity, coupled with the employment of state-of-the-art high-definition instruments alongside radiological expertise, serves to explain a margin of 0.2 mm discrepancy (which, notably, constitutes 10% of a 2 mm DOI) and should not be considered astonishing. Furthermore, an identical result was acheieved by Takamura and colleagues (line 276, Takamura, M.; Kobayashi, T.; Nikkuni, Y.; Katsura, K.; Yamazaki, M.; Maruyama, S.; Tanuma, J. ichi; Hayashi, T. A Comparative Study between CT, MRI, and Intraoral US for the Evaluation of the Depth of Invasion in Early Stage (T1/T2) Tongue Squamous Cell Carcinoma. Oral Radiol. 2022, 38, 114–125, doi:10.1007/s11282-021-00533-7). However, it's crucial to highlight that the measurement of pathological DOI is an objective metric, impervious to individual interpretation.
Regarding sample size, as we highlighted both in the title and the discussion, our study has to be considered a preliminary experience. We are aware that the sample is small, as we stated in our discussion, and that results need to be validated in a larger sample. As our work does not intend to ascertain differences between two treatment groups or two methods of measurement, there is no necessity for the computation of a sample. In summary, while determining the appropriate sample size remains a crucial element in any study, its determination should be guided by the nature of the research question and the design of the study.
Regarding ultrasound technique, both superficial and deeper lesions were scanned with the same technique: a small amount of ultrasound gel was placed into the cover used to coat the probe. In this way, a layer of gel created a distance between the probe and the mucosa. This method enabled to correctly delineate the morphology of the lesion (i.e. exophytic vs ulcerated). Text has been changed to clarify this point. Moreover, as the word “directly” (line 297) can be misunderstood, it was eliminated.
Both the papers by Izzeti and Di Stasio are now incorporated in the bibliography.
Thank you once again for your valuable input and guidance.
Kind regards.
Reviewer 2 Report
Dear Authors,
Please improve and extent your introduction
Please add more conclusion to your article
Kind regards
Minor editing of English language required
Author Response
Dear Reviewer,
Thank you for your kind revision. The requested changes have been successfully incorporated.
Kind regards.
Reviewer 3 Report
Study design is good but some grammatical error are there as well as self citation should be checked
spelling mistakes should be corrected
Author Response
Dear Reviewer,
Thank you for your kind revision. The grammatical errors and misspelling were checked and corrected. Regarding references, we cited some articles by our group because this paper has to be considered a natural continuation of what we reported in previous publications. Game-changer papers were also cited (i.e. Shintani, S.; Yoshihama, Y.; Ueyama, Y.; Terakado, N.; Kamei, S.; Fijimoto, Y.; Hasegawa, Y.; Matsuura, H.; Matsumura, T. The Usefulness of Intraoral Ultrasonography in the Evaluation of Oral Cancer. Int. J. Oral Maxillofac. Surg. 2001, 30, 139–143, doi:10.1054/ijom.2000.0035, Rocchetti, F.; Tenore, G.; Montori, A.; Cassoni, A.; Cantisani, V.; Di Segni, M.; Di Gioia, C.R.T.; Carletti, R.; Valentini, V.; Polimeni, A.; et al. Preoperative Evaluation of Tumor Depth of Invasion in Oral Squamous Cell Carcinoma with Intraoral Ultrasonography: A Retrospective Study. Oral Surg. Oral Med. Oral Pathol. Oral Radiol. 2021, 131, 130–138, doi:10.1016/j.oooo.2020.07.003, Takamura, M.; Kobayashi, T.; Nikkuni, Y.; Katsura, K.; Yamazaki, M.; Maruyama, S.; Tanuma, J. ichi; Hayashi, T. A Comparative Study between CT, MRI, and Intraoral US for the Evaluation of the Depth of Invasion in Early Stage (T1/T2) Tongue Squamous Cell Carcinoma. Oral Radiol. 2022, 38, 114–125, doi:10.1007/s11282-021-00533-7). However, in this new version of the manuscript, references regarding intraoral ultrasound and DOI were implemented. We added:
- Yesuratnam A, Wiesenfeld D, Tsui A, Iseli TA, Hoorn SV, Ang MT, et al. Preoperative evaluation of oral tongue squamous cell carcinoma with intraoral ultrasound and magnetic resonance imaging - Comparison with histopathological tumour thickness and accuracy in guiding patient management. Int J Oral Maxillofac Surg. (2014) 43:787–94. doi: 10.1016/j.ijom.2013.12.009
- Moreno KF, Cornelius RS, Lucas FV, Meinzen-Derr J, Patil YJ. Using 3 Tesla magnetic resonance imaging in the pre-operative evaluation of tongue carcinoma. J Laryngol Otol. (2017) 131:793–800. doi: 10.1017/S0022215117001360
- Goel V, Parihar PS, Parihar A, Goel AK, Waghwani K, Gupta R, et al. Accuracy of MRI in prediction of tumour thickness and nodal stage in oral tongue and gingivobuccal cancer with clinical correlation and staging. J Clin Diagnostic Res. (2016) 10:TC01–5. doi: 10.7860/JCDR/2016/ 17411.7905
- Chen WL, Su CC, Chen CM, Lee MC, Chen HC, Chen MK. MRI- derived tumor thickness: an important predictor of outcome for T4a- staged tongue carcinoma. Eur Arch Oto-Rhino-Laryngol. (2012) 269:959–63. doi: 10.1007/s00405-011-1685-9.
- Chammas MC, MacEdo TAA, Moyses RA, Gerhard R, Durazzo MD, Cernea CR, et al. Relationship between the appearance of tongue carcinoma on intraoral ultrasonography and neck metastasis. Oral Radiol. (2011) 27:1–7. doi: 10.1007/s11282-010-0051-8
- Lodder WL, Teertstra HJ, Tan IB, Pameijer FA, Smeele LE, Van Velthuysen MLF, et al. Tumour thickness in oral cancer using an intra-oral ultrasound probe. Eur Radiol. (2011) 21:98–106. doi: 10.1007/s00330-010-1891-7 Kodama M, Khanal A, Habu M, Iwanaga K, Yoshioka I, Tanaka T, et al. Ultrasonography for intraoperative determination of tumor thickness and resection margin in tongue carcinomas. J Oral Maxillofac Surg. (2010) 68:1746–52. doi: 10.1016/j.joms.2009.07.110
- Mark Taylor S, Drover C, MacEachern R, Bullock M, Hart R, Psooy B, et al. Is preoperative ultrasonography accurate in measuring tumor thickness and predicting the incidence of cervical metastasis in oral cancer? Oral Oncol. (2010) 46:38–41. doi: 10.1016/j.oraloncology.2009.10.005
- Kaneoya A, Hasegawa S, Tanaka Y, Omura K. Quantitative analysis of invasive front in tongue cancer using ultrasonography. J Oral Maxillofac Surg. (2009) 67:40–6. doi: 10.1016/j.joms.2007.08.006
- Baek CH, Son YI, Jeong HS, Chung MK, Park KN, Ko YH, et al. Intraoral sonography-assisted resection of T1-2 tongue cancer for adequate deep resection. Otolaryngol - Head Neck Surg. (2008) 139:805–10. doi: 10.1016/j.otohns.2008.09.017
- Yamane M, Ishii J, Izumo T, Nagasawa T, Amagasa T. Noninvasive quantitative assessment of oral tongue cancer by intraoral ultrasonography. Head Neck. (2007) 29:307–14. doi: 10.1002/hed.20523
- Songra AK, Ng SY, Farthing P, Hutchison IL, Bradley PF. Observation of tumour thickness and resection margin at surgical excision of primary oral squamous cell carcinoma - Assessment by ultrasound. Int J Oral Maxillofac Surg. (2006) 35:324–31. doi: 10.1016/j.ijom.2005.07.019
- Izzetti R, Nisi M, Gennai S, Oranges T, Crocetti L, Caramella D, Graziani F. Evaluation of Depth of Invasion in Oral Squamous Cell Carcinoma with Ultra-High Frequency Ultrasound: A Preliminary Study. Applied Sciences. 2021;11(16):7647. https://doi.org/10.3390/app11167647
Kind regards.
Round 2
Reviewer 1 Report
Dear authors,
You followed all the advice of all the reviewers. In this form, the article can be published.
Best regards